# Are There Any Parameters Missing in the Mathematical Models Applied in the Process of Spreading COVID-19?

**DOI:** 10.3390/biology10020165

**Published:** 2021-02-19

**Authors:** Pietro M. Boselli, Massimo Basagni, Jose M. Soriano

**Affiliations:** 1Group of Nutritional Modeling Biology, Department of Biosciences, University of Milan, 20133 Milan, Italy; piebosel@alice.it; 2Group of Biomathematics, Department of Biosciences, University of Milan, 20133 Milan, Italy; Basa.massimo@gmail.com; 3Department of Preventive Medicine and Public Health, Faculty of Pharmacy, University of Valencia, Burjassot, 46100 Valencia, Spain; 4Food & Health Lab, Institute of Materials Science, University of Valencia, Paterna, 46980 Valencia, Spain; 5Joint Research Unit on Endocrinology, Nutrition and Clinical Dietetics, University of Valencia-Health Research Institute La Fe, 46026 Valencia, Spain

**Keywords:** COVID-19, modelling study, Italy, Spain, UK

## Abstract

**Simple Summary:**

Nowadays, enhancing development of mathematical models is very important to help in the prediction of coronavirus disease 2019 (COVID1-19). However, the vast majority of published model-based predictions do not cover people who left the epidemic COVID-19 positive (alive) and they must be included in studies to guarantee a more accurate model for application in public health. The epidemic development phenomenon can be obtained with a modelling framework.

**Abstract:**

On 11 March 2020, coronavirus disease 2019 (COVID-19) was declared a pandemic by the World Health Organization (WHO). As of 12.44 GMT on 15 January 2021, it has produced 93,640,296 cases and 2,004,984 deaths. The use of mathematical modelling was applied in Italy, Spain, and UK to help in the prediction of this pandemic. We used equations from general and reduced logistic models to describe the epidemic development phenomenon and the trend over time. We extracted this information from the Italian Ministry of Health, the Spanish Ministry of Health, Consumer Affairs, and Social Welfare, and the UK Statistics Authority from 3 February to 30 April 2020. We estimated that, from the seriousness of the phenomenon, the consequent pathology, and the lethal outcomes, the COVID-19 trend relate to the same classic laws that govern epidemics and their evolution. The curve d(t) helps to obtain information on the duration of the epidemic phenomenon, as its evolution is related to the efficiency and timeliness of the system, control, diagnosis, and treatment. In fact, the analysis of this curve, after acquiring the data of the first three weeks, also favors the advantage to formulate forecast hypotheses on the progress of the epidemic.

## 1. Introduction

An outbreak of a novel coronavirus disease (COVID-19) was reported in December 2019 from Wuhan, China [1]. It was caused by a β-coronavirus (related to the severe acute respiratory syndrome coronavirus (SARS-CoV) [2] and the Middle East respiratory syndrome coronavirus (MERS-CoV) [3]) and called SARS-CoV-2 [4]. The World Health Organization (WHO) [5] declared this new disease a pandemic on March 11, 2020. To date, SARS-CoV-2 infected patients have developed in four ways: (i) no symptoms, (ii) mild/moderate, (iii) severe, and iv) critical symptoms. For mild symptoms, patients reported dry cough, sore throat, and fever. Meanwhile, the population with severe symptoms had pneumonia and shortness of breath; while those with critical symptoms were characterized as having respiratory failure, septic shock, and multi-organ failure [6]. It was suggested that mortality was correlated with health-care burden [7].

Originally, mortality rate (based on the number of deaths relative to the number of confirmed cases of infection) was 3.6% and 1.5% from COVID-19 inside and outside of China, respectively [8]. Baud et al. [9] re-estimated mortality rates (dividing the number of deaths on a given day by the number of patients with confirmed COVID-19 infection 14 days before, as that was the maximum incubation period) [10], obtaining 5.6% and 15.2% inside and outside of China, respectively. The measurement of the isolation of cases and contacts [11] and model-based predictions are useful to aid policy makers to opportunely carry out the right decisions [12] and take control of this pandemic [13]. Several model-based predictions have been published for this disease in relation to the case fatality ratio [14], the reproductive number of the virus to make predictions of daily new cases on ships [15], spreading dynamics of the emerged coronavirus epidemic in China for a three-week forecast [16], travel restrictions [17], potential for sustained human-to-human transmission to occur in locations outside Wuhan if cases were introduced [18], and non-pharmaceutical interventions and quarantine [19,20].

The aims of this study were to describe the process of the spread of COVID-19 from February to April 2020 in Italy, Spain, and UK using mathematical logistic models, and to propose a model to follow the evolution of COVID-19 in an isolated population and predict its duration.

## 2. Materials and Methods

The official information was obtained from the Italian Ministry of Health [21], the Spanish Ministry of Health, Consumer Affairs, and Social Welfare [22], and the UK Statistics Authority [23]. We extracted the total number of people infected, recovered, and deaths of COVID-19 from February 3 to 30 April 2020, and stored these on Excel files. The observed data indicated that the epidemiological process can be considered "normal". In fact, during the first three weeks, COVID-19 seemed to evolve according to the most common epidemic lines. We decided to implement logistic-normal model, using variables that represent the cumulative values of: 

a: total number of dead people at time t.

b: total number of recovered people at time t.

c: a + b; sum of the total number of people who left the epidemic at time t.

d: e − c; number of COVID-19 positive (alive) patients who are still in the epidemic at time t.

In accordance with the theoretical considerations and clinical data, e(t), a(t), b(t), and c(t) have the typical form of logistic functions. These values start from zero at the beginning and reach their maximum value asymptotically when the epidemic slows down until it disappears.

On the other hand, the function d(t) seems to have the form of a second-order function, in accordance with the “in–out” scheme. In fact, the number of people who remain in the epidemic at recovered time, t, correspond to the difference between the number of people who entered (e) and those who left (c) the epidemic.

The collected data represents the total number of officially registered cases. They therefore include epidemiological data from various regions and different areas, both from the point of view of economic and social parameters and climatic characteristics in the season considered. These characteristics make the sample sufficiently representative of a complex reality and, therefore, conditions are quite inclusive of what the epidemic phenomenon could encounter during its evolution.

The total observation period, up to the moment of the drafting of these notes (about 3 months), could be considered sufficient for the research of the main characteristics of the epidemic, and could be also used to attempt the construction of a formal mathematical model, useful not only for carrying out the description of the epidemiological phenomenon, but also for the forecast estimates of the subsequent evolution and the terms of exit from the current pandemic phase. A heuristic model of logistic functions was implemented, and MS Excel version 14.0 from Microsoft Excel^®^ (Microsoft Inc. Redmond, WA, USA) was used to analyze the data.

## 3. Results

### 3.1. Mathematical Modelling

Nowadays, the formal models used are based on the classical theory of population dynamics as regards functions: e(t), a(t), b(t). The experimental values collected up to day t* (where t* = total number of observation days, which began on 29 February and ended on 30 April 2020) allowed fitting directly from these values with a logistic model.

The only arbitrary data entered was the asymptotic value of the function and **e**(t); (**e**_max_ = Final-asymptotic total positives) that is, the hypothetical estimate of the maximum number of patients who will “enter” as infected within a certain time (where t** is defined as end-epidemic). This **e**_max_ value can be hypothesized (assigned), based on the observation of the trend of the curve and **e**(t) and on considerations relating to the measures to contrast the epidemic, but remains a priori within certain arbitrary limits.

The asymptotic values of the other two functions, **a**(t) and **b**(t), were assigned, starting from the estimates of the final percentages of the dead (**a**) and recovered (**b**) subjects from the total of the infected patients. The percentages of a(t) were fairly stabilized from a certain time onwards: a fairly severe asymptotic value (**a**_max_) was assumed as a hypothesis, based on the experimental values observed, as follows:

**a**_max_ = 26,675/197,600 = 13.5% (estimate from 26 April 2020 for Italy).

**a**_max_ = 23,190/215,000 = 10.8% (estimate from 26 April 2020 for Spain).

**a**_max_ = 26,771/120,614 = 22.2% (estimate from 30 April 2020 for UK).

The estimate of t*** (the time it takes for the b(t) curve to reach its asymptotic) follows accordingly, on the basis of general knowledge and the resolution time of the disease (about four weeks on average). This data allows us to hypothesize that the asymptotic value, bmax = emax, is reached at time t*** >= t**.The following formulas constitute the model adopted for the description of the epidemic development phenomenon and for the calculations of the functions that represent the trend over time. We want to emphasize once again that the inserted arbitrary hypothesis concerns the value of the asymptote of the curve **e**(t), which was assigned on the basis of experiences related to similar events studied in the past and the recent exchange of points seen with some epidemiologists, and the General and Reduced Logistic Model were developed.

For the General Logistic Model, we used two logistic models; (i) Verhulst [24] described the growth of a biological population, and (ii) the Lotka’s model [25] called the “Law of Population Growth”. According to these models, the general logistic model in our study is the following:

L(t) = L_max_ /(1 + q e^−K(t−t_o_^^)^),

where: 

L(t) is the numerical value that the logistic equation returns at time t.

L_max_ is the asymptotic value that logistics take for t tending to infinity.

L_0_ is the numeric value that the equation returns at the beginning (for t = 0).

q = (L_max_ − L_0_)/L_0_.

K represents the growth rate of L (slope of the curve).

t is called the current observation time.

t_0_ is called the initial observation time.

Outbreak Start: (e(t_o_) = c(t_o_) = 0), t_o_ = 0.

L_e_/(1 + q_e_ e^−K^_e_^(t−t_o_^^)^) = L_c_ /(1 + q_c_ e^−Kc(t−t_o_^^)^).

L_e_/(1 + q_e_) = Lc/(1 + q_c_).

L_e_/L_c_ = (1 + q_e_)/(1 + q_c_).

End of Epidemic: c(t) = e(t): L_c_/(1 + qc e^-Kc(t−t_o_^^)^) = L_e_/(1 + q_e_ e^−Ke(t−t_o_^^)^).

L_c_/(1 + 0) = L_e_/(1 + 0).

L_c_ = L_e_.

For the Reduced Logistic Model, although theoretically less “fit” with the previous one with the experimental data, it allows the estimate of parameters in a simplified way. The difference compared to the model described above consists of the assumption that the inflection point of the curve is located exactly at 1/2 of the asymptotic value.

Simplified model equation:

d(t) = **e**(t) − **c**(t) = L_e_/(1 + e^−Ke(t−t_o_^^)^) − L_c_ /(1 + e^−Kc(t−t_o_^^)^);

End of epidemic: **d**(t) = **e**(t) = **c**(t) = 0;

e(t) = L_e_/(1+e ^(0)^) = 1/2L_e_;

c(t) = L_c_(1+e ^(0)^) = 1/2L_c_;

where t_o_ = time of the observed inflection point (in respect to **e**(t) and **c**(t)), L_i_ = asymptotic value of the curve (assigned according to the data collected), and K_i_ = growth rate (inclination of the curve) at the inflection (estimated from the data collected).

### 3.2. Results for COVID-19 in Italy, Spain, and UK 

The results, obtained by analyzing the data, conducted with the use of the models described above, are shown in Figure 1, Figure 2 and Figure 3 for Italy, Spain, and UK, respectively. The mathematical models adopted interpret the epidemic phenomenon in all three countries, although for the United Kingdom, the observed data are limited to the total number of the infected and dead.

The graphs show the trends over time, that is, the evolution of the pandemic, according to a hypothesis of medium severity, even in the four weeks following the last recording of the observed data.

As for the epidemic, it has been hypothesized that:

e(30 April 2020) = 251,800 total infected in Italy (Figure 1);

e(30 April 2020) = 235,200 total infected in Spain (Figure 2);

e(30 April 2020) = 162,200 total infected in the UK (Figure 3).

It should be noted that the characteristic curve of the trend of the fundamental function d(t) of the COVID-19 positive (alive) population who are still in the epidemic is subject to more contained, if not negligible, variations. For this reason, among all the functions, it is more convenient to use the d(t) to understand the progress of the epidemic and hypothesize its duration. In addition, the function d(t) can indicate whether the epidemiological situation, which has arisen, confirms or denies the effectiveness of the containment measures already adopted in the area. The theoretical curve of positive (alive) patients was found by the least squares method, by varying the parameters of entry and output logistic functions to minimize the total deviance of the system.

The terminal part of the curve d(t) tends to its asymptote (zero), with a time delay with respect to what happens for the functions e(t) and c(t). This data is consistent with the fact that the pathology requires a time delay to resolve itself, regardless of how it evolves. The slight asymmetry of the curve d(t) is therefore evident. The main cause of the asymmetry lies in the fact that, while at the beginning, the entry into the disease was immediate and overwhelming, the exit was slower, because in d(t) there are still those who entered a few weeks earlier. It was also evident that less control of the curve e(t) would lead to a dangerous turnaround and to the further postponement of the end of the epidemic. Other causes of this asymmetry are the following: (i) Components of the epidemic phenomenon are not synchronous.; contagion occurs after the disease, followed by hospitalization, and finally, they recovered or died. (ii) Human behavior can produce a change in the epidemic pattern. (iii) Errors in health surveillance services (change in evaluation criteria, reporting, collection, and transmission of data) can also lead to changes in the trends of the curves.

Figure 4, Figure 5 and Figure 6 reflects the COVID-19 epidemic time-courses during the second wave in Italy, Spain, and UK, respectively. The transition period from the contagion (entry) to recovered status (exit)/death (exit) can still spread the infection. This means that, while positive (alive) patients exist, the epidemic continues over time. The positive (alive) patients, d(t) = e(t) − b(t) − a(t), develop with a “bell” trend, which starts from zero, rises to a maximum, and then decreases to zero again. An agreement can be said to exist where the duration of the epidemic coincides with the duration of the d(t).

To illustrate, Figure 7 shows, for Italy, two predictive situations based on ineffective (red triangles above the yellow line) and effective (green triangles below the yellow line) anti-COVID-19 measures. This last will happen when the population uses adequate personal protective equipment, social distancing, and COVID-19 vaccination during the COVID-19 pandemic.

## 4. Discussion

To date, all published mathematical modelling studies of COVID-19 take account of the online official sources for each country, for the number of deaths, recovered, and infected. Our results demonstrated that, despite the seriousness of the phenomenon, the consequent pathology, and the lethal outcomes, the trends of COVID-19 are governed by the same classic laws that govern epidemics and their evolution. In fact, the curve d(t) can give useful information on the duration of the epidemic phenomenon. Since d(t) = e(t) − c(t) = e(t) − b(t) − a(t), it is obvious that the more you can increase the number of infected and/or decrease the number of recovered and dead, the longer the epidemic lasts. The evolution of the epidemic depends, above all, on the efficiency and timeliness of the system, control, diagnosis, and treatment. The analysis of the curve d(t) also has the advantage of allowing, after acquiring the data for the first three weeks, the formulation of forecast hypotheses on the progress of the epidemic. These hypotheses are indispensable in the event that the health system has been caught by surprise, to help reorganize the control, diagnosis, and treatment measures. For this reason, the use of model-based predictions can help with the understanding of this disease and its progression, as and when the ongoing data collection and epidemiological analyses are made. At first, the WHO [26] had estimated, in January 2020, that R0 (or Basic Reproduction Number) for COVID-19 was of 1.4-2.5. However, this was reviewed by Imperial College London [27], who estimated R0 with a range of between 1.5 and 3.5. On the Diamond Princess cruise ship, in which 355 out of 3711 people were confirmed as having COVID-19 infection, Zhang et al. [15] carried out an estimation of the reproductive number (R0) of the illness with a value of 2.28 during the early stage; R0 values of 2.20 and 2.68 were obtained in Wuhan [28] and five provinces of China (Hubei, Guangdong, Henan, Zhejiang, and Hunan) [20], respectively. The variability of these values is reflected in the assumption of different incubation periods, grade of infectiousness, and estimated asymptomatic patients. Park et al. [29] carried out a systematic review of the published literature and preprints on COVID-19, detecting several methods such as hierarchical, exponential growth, epidemic growth, stochastic simulations of outbreak trajectories, analysis of epidemiological data, susceptible-exposed-infectious-recovered models, among others. In accordance with Jahedi and Yorke [30], we agree that best pandemic models are the simplest, and our study took into account the data of the first three weeks, focusing on effectively evaluating the progress of the epidemic. Furthermore, other logistic models have been applied to coronavirus, i.e., Ahmed et al. [31] studied mathematical models in forecasting epidemic size for Turkey and Iraq; Cotta et al. [32] applied a SIRU-type epidemic model for the prediction of the COVID-19 epidemic evolution in Brazil; whilst Tang et al. developed a model combined with intervention, such as treatment, isolation (hospitalization), and quarantine [33], and expanded with a time-dependent dynamic system [34].

## 5. Conclusions

In conclusion, the practical usefulness of this approach as well as the theoretical interest has been demonstrated. The two new values, c and d, are the sum of the total number of people who left the epidemic at time t and number of COVID-19 positive (alive) patients who are still in the epidemic at time t, respectively. It is valid as a tool for an integral understanding and forecasting hypothesis of the epidemic phenomenon. After all, the more we know about this virus and disease, the easier it will be to beat it.

## Figures and Tables

**Figure 1 biology-10-00165-f001:**
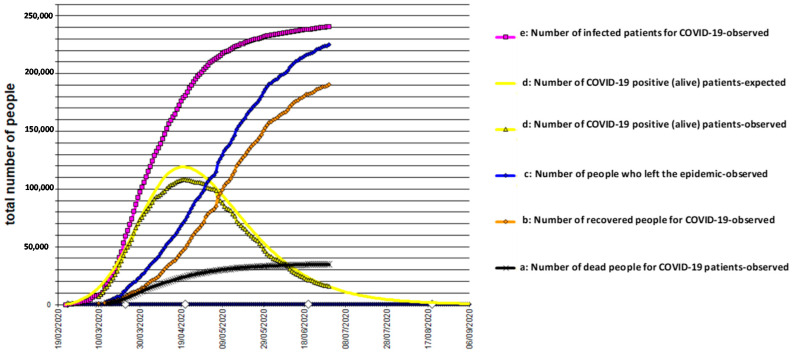
COVID-19 epidemic time-course in Italy.

**Figure 2 biology-10-00165-f002:**
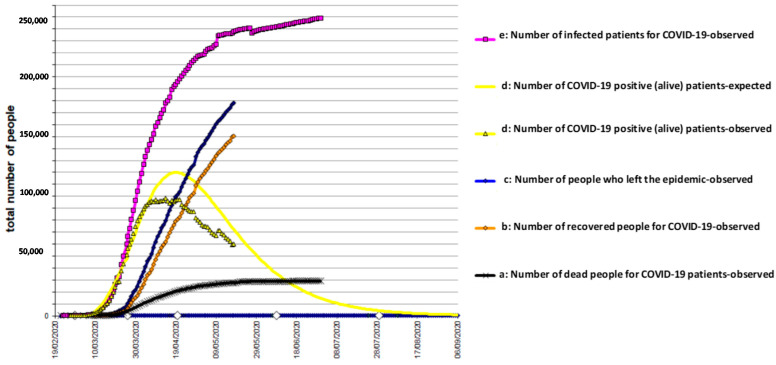
COVID-19 epidemic time-course in Spain.

**Figure 3 biology-10-00165-f003:**
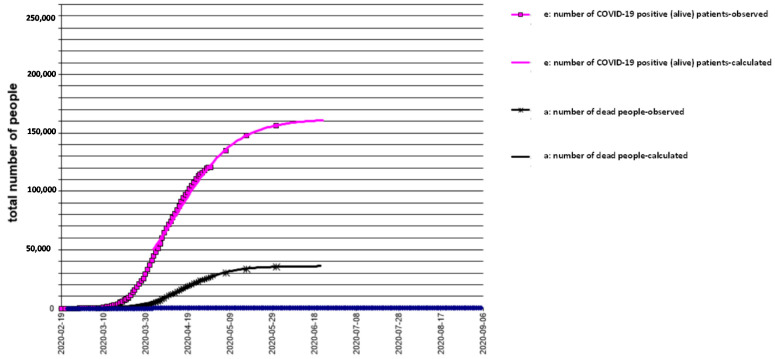
COVID-19 epidemic time-course in UK.

**Figure 4 biology-10-00165-f004:**
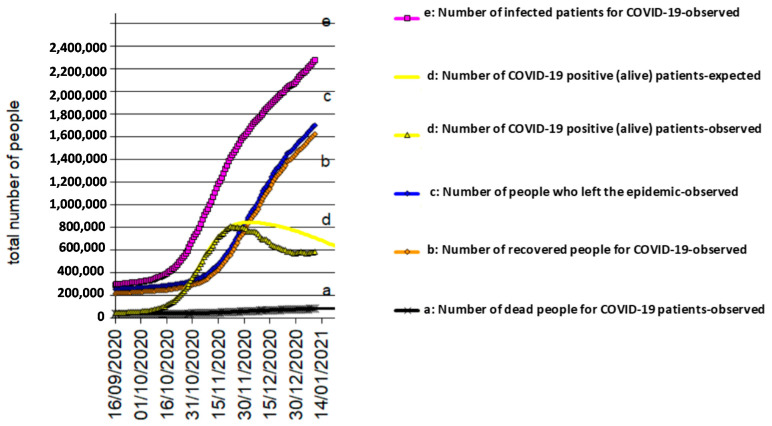
COVID-19 epidemic time-course in Italy during the second wave.

**Figure 5 biology-10-00165-f005:**
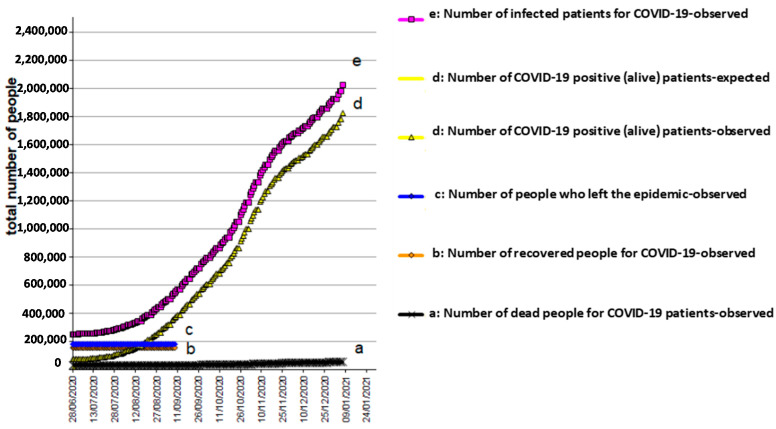
COVID-19 epidemic time-course in Spain during the second wave.

**Figure 6 biology-10-00165-f006:**
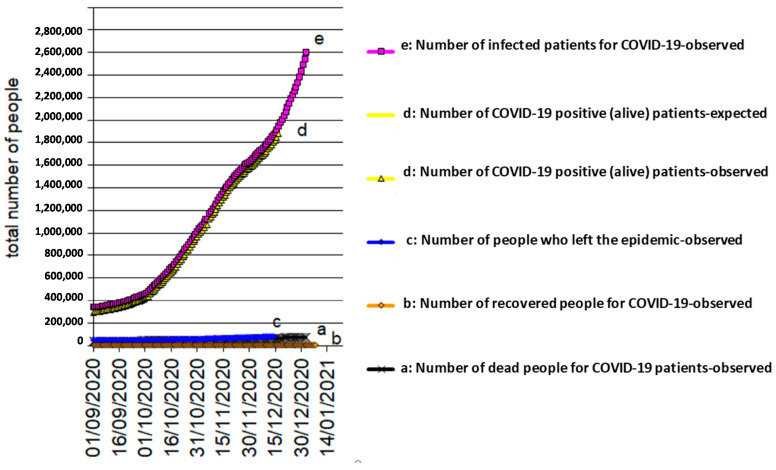
COVID-19 epidemic time-course in UK during the second wave.

**Figure 7 biology-10-00165-f007:**
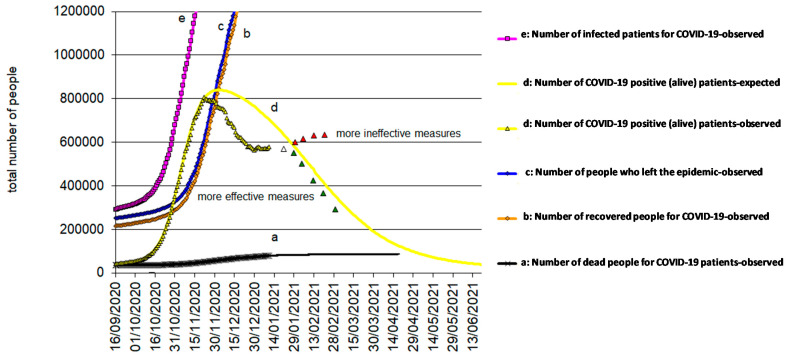
COVID-19 epidemic time-course in UK during the second wave, in detail, and incidence of this disease with effective and ineffective measures.

## Data Availability

The authors confirm that the data supporting the findings of this study are available in publicly accessible repositories within the bibliography.

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
