# Peer review of "Are There Any Parameters Missing in the Mathematical Models Applied in the Process of Spreading COVID-19?"

_biology, 2021, doi:10.3390/biology10020165_

Round 1

Reviewer 1 Report

No additional recommendations

Reviewer 2 Report

The changes have been applied.

This manuscript is a resubmission of an earlier submission. The following is a list of the peer review reports and author responses from that submission.

Round 1

Reviewer 1 Report

This is a study to describe the process of spreading COVID-19, from February to April, 2020, ininItaly, Spain and the UK with mathematical models. This study proposes a model to follow the evolution of COVID-19 in an isolated population and predict its duration.

I would suggest rejection: The content of the article is not argued in a coherent way, it is in fact not in line with the question posed in the title. There is confusion in the development of the contents that does not allow us to understand the essence of reasoning and study.

  1. The question posed in the title is original but the answer given in the article is not well defined. It is not clear what progress has been made with respect to current knowledge.
  2. The article is not written properly:
    1. Sentences that are too long should be simplified
    2. The introduction does not deal with the background regarding the mathematical models used in the study of covid-19 . It also does not report previous studies using logistic models
    3. Equations should be explained.
    4. Variables should be explicitly explained
      1. The curve d(t) it is not clear what it refers to (line 31 Abstract)
      2. Variable e(t) at line 8
  3. The methods, the tools, the software are not sufficiently described in detail
  4. the code is not public
  5. no reference is made to the software used
  6. Ssentencesin English are not very appropriate and are sometimes incomprehensible
  7. thefigures have low resolution and hardly legible labels

Author Response

Reviewer 1

Reviewer’s comment: This is a study to describe the process of spreading COVID-19, from February to April, 2020, ininItaly, Spain and the UK with mathematical models. This study proposes a model to follow the evolution of COVID-19 in an isolated population and predict its duration. I would suggest rejection: The content of the article is not argued in a coherent way, it is in fact not in line with the question posed in the title. There is confusion in the development of the contents that does not allow us to understand the essence of reasoning and study.

Author’s comment: The purpose of the work is to suggest a way to predict the duration of an epidemic/pandemic. It is obvious that you enter the epidemic through contagion and only go out or recovered or die. Therefore, the epidemic/pandemic finish until all infected are recovered or dead. Therefore, the trend of the difference between the number of infected and the sum of recovered +dead allows to estimate the duration of the epidemic/pandemic. This difference defines the category of "positive-alive" (reflected as curve d). Following the trend of positive-living represents a real progress compared to current epidemiological knowledge.

Reviewer’s comment: The question posed in the title is original but the answer given in the article is not well defined. It is not clear what progress has been made with respect to current knowledge.

Author’s comment: According to this comment, we have completed the last paragraph, in discussion section, to explain the progress made with respect to current knowledge.

Reviewer’s comment: The article is not written properly: Sentences that are too long should be simplified

Author’s comment: Manuscript has been reviewed and reduced sentences that were too long.

Reviewer’s comment: The introduction does not deal with the background regarding the mathematical models used in the study of covid-19 . It also does not report previous studies using logistic models

Author’s comment: Mathematics provides different models for the study of biological phenomena. They are mainly compartmentalized and exponential. As for viruses and bacteria, "logistic" functions are commonly used. These, better than any other function, describe the phenomenon of diffusion until a maximum value of "saturation" is reached.

In this work the total number of infected (curve c), the total number of deaths (curve a), the total number of recovered people (curve b), the total number of recovered and dead (curve c) are described by logistic functions. The number of people remaining in the epidemic (positive-live, curve d) is given, on the other hand, by the difference of two logistic functions (e-c). Thus, the positive-live curve is the novelty compared to the models used so far. It has a bell pattern, not saturating. It starts from scratch, reaches a maximum when the difference between the curve of total infections (e) and that of recovered+dead (c) is maximum. Return to zero only when the number of recovered+dead equals the number of infected. For this reason, the analysis of the trend of curve d allows to estimate the duration of the epidemic/pandemic. Only when curve d is zeroed, the epidemic/pandemic can declare itself over.

Reviewer’s comment: Equations should be explained. Variables should be explicitly explained.

Author’s comment: According to this comment, we have explained in the manuscript.

In fact, for General Logistic Model, Pierre F. Verhulst in 1838 published the logistic model to describe the growth of a biological population (see Verhulst, P.F., 1838. Notice sur la loi que la population suit dans son accroissement. Correspondance Math. Phys. 10, 113–121). It was taken over by Alfred J. Lotka in 1925 and called the "Law of Population Growth" (Lotka, A. J. 1925. Elements of physical biology. – Williams and Wilkins). On the other hand, general equation of the model:

L(t) = Lmax /(1 + q e-K(t-to))       

Where:

L(t) is the numerical value that the logistic equation returns at time t

Lmax is the asymptotic value that logistics take for t tending to infinity

L0 is the numeric value that the equation returns at the beginning (for t=0)

q = (Lmax - L0)/L0

K represents the growth rate of L (slope of the curve)

t is called the current observation time

t0 is called the initial observation time

Outbreak Start: (e(to)=c(to)=0), to=0

Le /(1 + qe e-Ke(t-to)) = Lc /(1 + qc e-Kc(t-to))       

Le /(1 + qe) = Lc /(1 + qc)       

Le / Lc = (1 + qe)/(1 + qc)       

End of  Epidemic: c(t)=e(t): Lc /(1 + qc e-Kc(t-to)) = Le /(1 + qe e-Ke(t-to))

Lc /(1 + 0) = Le /(1 + 0)

Lc = Le

Reviewer’s comment: The curve d(t) it is not clear what it refers to (line 31 Abstract).

Author’s comment: The curve d (t) is the positive alive, whose tendency is important to hypothesize the duration of the epidemic.

Reviewer’s comment: Variable e(t) at line 8

Author’s comment: The e(t) is not a variable; it is the curve of determined total infections. It has been changed in the results section.

Reviewer’s comment: The methods, the tools, the software are not sufficiently described in detail and the code is not public

Author’s comment: According to this comment, methods, tools and software have been carried out to extensive description. The official information is public  and is obtained from the Italian Ministry of Health [21], the Spanish Ministry of Health, Consumer Affairs and Social Welfare [22] and the UK Statis-tics Authority [23] and MS Excel version 14.0 (Microsoft Corporation) was used to analyze the data.

Reviewer’s comment: no reference is made to the software used

Author’s comment: Data processing was carried out using the mathematical/statistical functions made available by Microsoft Excel.

Reviewer’s comment: Sentences in English are not very appropriate and are sometimes incomprehensible

Author’s comment: According to this comment, we have reviewed the manuscript to improve this comment.

Reviewer’s comment: The figures have low resolution and hardly legible labels

Author’s comment: According to this comment, figures have been improved.

Reviewer 2 Report

Dear Authors Thank you very much for your submission. Please clarify these points: 1- What are the references of your models 2-please clearly explain the models and their usage in the epidemic prediction 3-please find the papers that been previously used these models and their conclusions 4-Please expand your discussion, 5-Please explain what is the main conclusion of your study, it is not clear thank you very much

Author Response

Reviewer 2

Reviewer’s comment: Dear Authors Thank you very much for your submission. Please clarify these points: 1- What are the references of your models

Author’s comment: For General Logistic Model, Pierre F. Verhulst in 1838 published the logistic model to describe the growth of a biological population (see Verhulst, P.F., 1838. Notice sur la loi que la population suit dans son accroissement. Correspondance Math. Phys. 10, 113–121). It was taken over by Alfred J. Lotka in 1925 and called the "Law of Population Growth" (Lotka, A. J. 1925. Elements of physical biology. – Williams and Wilkins). On the other hand, general equation of the model:

L(t) = Lmax /(1 + q e-K(t-to))       

Where:

L(t) is the numerical value that the logistic equation returns at time t

Lmax is the asymptotic value that logistics take for t tending to infinity

L0 is the numeric value that the equation returns at the beginning (for t=0)

q = (Lmax - L0)/L0

K represents the growth rate of L (slope of the curve)

t is called the current observation time

t0 is called the initial observation time

Outbreak Start: (e(to)=c(to)=0), to=0

Le /(1 + qe e-Ke(t-to)) = Lc /(1 + qc e-Kc(t-to))       

Le /(1 + qe) = Lc /(1 + qc)       

Le / Lc = (1 + qe)/(1 + qc)       

End of  Epidemic: c(t)=e(t): Lc /(1 + qc e-Kc(t-to)) = Le /(1 + qe e-Ke(t-to))

Lc /(1 + 0) = Le /(1 + 0)

Lc = Le

According to this comment, we have added in the manuscript.

Reviewer’s comment: 2-please clearly explain the models and their usage in the epidemic prediction

Author’s comment: According to this comment, we have re-written the manuscript to explain the models and their usage in the epidemic prediction.

Reviewer’s comment: 3-please find the papers that been previously used these models and their conclusions

Author’s comment: According to this comment, we have added in the manuscript.

Reviewer’s comment: 4-Please expand your discussion,

Author’s comment: According to this comment, we have expanded the discussion section.

Reviewer’s comment: 5-Please explain what is the main conclusion of your study, it is not clear thank you very much

Author’s comment: According to this comment, we have re-written the main conclusion.

Reviewer 3 Report

See attached file

Author Response

I send you as attachment the author's notes to reviewer (See Reviewer3.docx below)

Round 2

Reviewer 2 Report

Thank you for the revision

Line 135 and 136 please replace the XX and XXX

Author Response

Thank you for the revision. Line 135 and 136 please replace the XX and XXX Author’s comment: Thank you for your comment. According to your comment, we have replaced XX and XXX for references number 24 and 25, respectively.

Reviewer 3 Report

You still have not stated how the models were fitted. Was it by nonlinear least squares or some other approach. Saying that you used Excel is inadequate without specifying the method.

Your discussion on page 6 recognizes that the shape of the epidemic is asymmetric but you do not consider any alternative approach to resolve the problem.

Author Response

Reviewer’s comment: You still have not stated how the models were fitted. Was it by nonlinear least squares or some other approach. Saying that you used Excel is inadequate without specifying the method. Author’s comment: According to your comment, we have added the model in materials and methods section, as follows: “”A heuristic model of logistic functions was implemented and MS Excel version 14.0 from Microsoft Excel® (Microsoft Inc. USA) was used to analyze the data”. Furthermore, we have added in results section this phrase to clarify it: “The theoretical curve of positive (alive) patients was found by the least squares method by varying the parameters of entry and output logistic functions to minimize the total deviance of the system.” In fact, the use of Excel in these procedures is usual in COVID-19 outbreaks: 1. Niazkar et al. (2020)1analyzed using Excel software, which provides suitable facilities for data analysis and numerical implementation, according to previous study (Niazkar and Afzali, 2016)2. 2. Chatterjee et al. (2020)3 modelled stochastically the method using MC simulations on MS Excel in COVID-19 epidemic applied in India. 3. De Noni et al. (2020)4 applied MS-Excel in a two-wave epidemiological model of COVID-19 outbreaks.